# Highs *and* lows: Genetic susceptibility to daily events

**Maurizio Sicorello**[1,2], **Linda Dieckmann**[1], **Dirk Moser**[1], **Vanessa Lux**[1], **Maike Luhmann**[3], **Andreas B. Neubauer**[4], **Wolff Schlotz**[5,6], **Robert Kumsta**[1] *

1 Department of Genetic Psychology, Faculty of Psychology, Ruhr-University Bochum, Germany, 2 Department of Psychosomatic Medicine and Psychotherapy, Central Institute of Mental Health, Medical Faculty Mannheim, Heidelberg University, Heidelberg, Germany, 3 Department of Psychological Methods, Faculty of Psychology, Ruhr-University Bochum, Bochum, Germany, 4 DIPF | Leibniz Institute for Research and Information in Education, Frankfurt am Main, Frankfurt, Germany, 5 Max-Planck-Institute for Empirical Aesthetics, Frankfurt am Main, Frankfurt, Germany, 6 Institute of Psychology, Goethe University, Frankfurt am Main, Frankfurt, Germany

* robert.kumsta@rub.de

**Data Availability Statement:** Data and analysis script to reproduce our results are available on the open science framework: https://osf.io/zpvns/?view_only=d1b01be029414512ad835db6192695cd.

## Abstract

Why people differ in their susceptibility to external events is essential to our understanding of personality, human development, and mental disorders. Genes explain a substantial portion of these differences. Specifically, genes influencing the serotonin system are hypothesized to be *differential susceptibility factors*, determining a person's reactivity to both positive *and* negative environments. We tested whether genetic variation in the serotonin transporter (*5-HTTLPR*) is a differential susceptibility factor for daily events. Participants (*N* = 326, 77% female, mean age = 25, range = 17–36) completed smartphone questionnaires four times a day over four to five days, measuring stressors, uplifts, positive and negative affect. Affect was predicted from environment valence in the previous hour on a within-person level using three-level autoregressive linear mixed models. The *5-HTTLPR* fulfilled all criteria of a differential susceptibility factor: Positive affect in carriers of the short allele (S) was less reactive to both uplifts and stressors, compared to homozygous carriers of the long allele (L/L). This pattern might reflect relative affective inflexibility in S-allele carriers. Our study provides insight into the serotonin system's general role in susceptibility and highlights the need to assess the whole spectrum of naturalistic experiences.

## Introduction

Experiences shape who we are, whether it is big breaks in our biography or minor cumulative events. However, people vary markedly in how strongly they are affected even by the same events. In the face of trauma, some are vulnerable to post-traumatic stress, while others are resilient. Some are exalted by personal achievements, compliments or novel sensations, while others remain even-minded. This variability poses a fundamental challenge to our understanding of personality, human development, and the etiology of mental disorders.

**Funding:** The author(s) received no specific funding for this work.

**Competing interests:** The authors have declared that no competing interests exist.

Differential susceptibility theory offers a framework to understand these individual differences [1] (for related for related theoretical accounts see also [2, 3]). While classic diathesis-stress models aim at explaining why some people are more *vulnerable* to negative events than others, the theory posits that some factors make people more *susceptible* to both positive and negative events. The theory's inception was largely motivated by evolutionary considerations [4]. Genes are the prototypical invariant traits to explain why people are differentially affected by environmental influences [5] and therefore represent a central aspect of personality and affective science [6].

For affective vulnerability, substantial research described its relationship to both genetics and personality: Affective reactivity to negative events appears to be largely inherited [7], relatively stable over time [8], and fundamentally connected to trait neuroticism [8–10]. Linking these components, there is evidence that the relationship between neuroticism and the variability of negative affect in everyday life is in part due to genetic influences [11]. However, these findings concern only the negative spectrum of experiences. The differential susceptibility perspective is less well researched, despite heritability coefficients simulated from differential susceptibility theory being compatible with heritability coefficients usually observed in twin studies [12].

Molecular genetics is integral to follow up the evidence from heritability studies by answering not only *if*, but *how* genes are related to specific aspects of personality [6]. Most importantly, understanding concrete genes related to the activity of specific neurotransmitters helps elucidate the role of larger scale neurobiological systems. The serotonin system has emerged as one of the most promising biological substrates of differential susceptibility [13]. Most serotonergic neurons originate from the raphe nuclei in the brainstem and project to many distinct brain regions [14]. Due to this combination of a regionally highly restricted origin and a large network of projections, the serotonin system is in an excellent position to mediate broad valence-general responses to external events, in favor of serotonin being a simple neural substrate of good-versus-bad mood as often claimed [13, 15]. Fittingly, one of the first potential susceptibility gene variants considered was the serotonin-transporter-linked polymorphic region (*5-HTTLPR*; [1]).

The *5-HTTLPR* originally became the poster-child for vulnerability genes when the Dunedin longitudinal study found that the short version of the gene (S) exacerbated the effect of child maltreatment on depression, compared to the long version (L) [16]. It consists of a 43 base pair insertion/deletion polymorphism in the promotor region of the serotonin transporter gene (SLC6A4) which influences the efficiency of serotonin reuptake from the synaptic cleft, with the L-variant leading to a more efficient reuptake through higher serotonin transporter gene transcription rates [17]. Unfortunately, replication studies and several meta-analyses produced conflicting results [18, 19], often attributed to false positives as a consequence of low statistical power [20]. Beyond depression, however, there is evidence that the *5-HTTLPR* can also increase the affective responses to positive environment [21] and several more process-oriented meta-analyses indicate that the *5-HTTLPR* plays a role in more basic physiological reactivity and affective functioning, even when publication bias is considered [22–24]. These robust effects corroborate that the *5-HTTLPR* is a viable candidate to test competitive theories concerning the function of the serotonin system.

The hypothesis that the *5-HTTLPR* might actually be a susceptibility factor, and not merely a vulnerability factor, was mainly conceived based on two observations [25]: First, most studies focused on the influence of adverse environments (e.g. maltreatment) on negative outcomes (e.g. depression), neglecting positive aspects on both sides of the equation (e.g. family coping resources and sense of coherence). Second, most studies did not further examine that carriers of the S-allele not only reported more depressive symptoms when they experienced many

severe stressors, but S-allele carriers also reported fewer depressive symptoms when they experienced very few stressors. Since then, formal statistical criteria emerged to differentiate between vulnerability and differential susceptibility [1, 26] (see Results section for details). Still, to date, the evidence is unclear as meta-analyses have not provided unequivocal evidence that the *5-HTTLPR* is a differential susceptibility factor. One meta-analysis found that negative developmental environments increased the likelihood of negative outcomes for carriers of the S-allele, compared to L/L-carriers, but did not find a robust overall effect of positive developmental environments [27]. Another meta-analysis based on randomized controlled experiments did not show a significant difference in differential susceptibility for the 5-HTTLPR [28]. Interestingly, they observed that the pooled effect size of different hypothesized susceptibility genes depended on the timescale of measurements, with the largest effects for interventions which focus on immediate neural or behavioral responses to negative or positive stimuli.

Inconsistent findings in this area might be partly due to the fact that the overwhelming majority of studies attained cross-sectional data, and correlations were usually computed on a *between-person* level (e.g. [18]). However, both differential susceptibility and the vulnerability to stressors are the property of a dynamic system that only manifests over time. Consequently, the *within-person* association of repeated measurements collected from the same person is better suited to capture differential susceptibility [29]. This association reflects how contingent a person's state is on immediate experiences (e.g. "when person A is in a more stressful environment, s/he feels more distressed"). Accordingly, we argue that if the differential susceptibility hypothesis is correct, the *5-HTTLPR* should moderate the individual strength of this correlation such that it is stronger for carriers of an S-allele than for carriers of only L-alleles.

Ecological momentary assessment (EMA) offers the opportunity to test differential susceptibility through person-specific models of environmental reactivity to real life experiences with high ecological validity. EMA methods are capable of measuring even minor daily experiences which often have a higher predictive utility than major life events and were the main target in the present study [30, 31]. Moreover, momentary assessments approximate biological processes more closely than classic retrospective and trait self-report measures [32].

So far, two EMA studies found a positive association between the S-allele and emotional reactivity, but only measured negative experiences and therefore could not test for differential susceptibility [33, 34]. In addition, their daily diary method with one daily retrospective assessment in the evening most likely captured only relatively persistent stress reactions, comes with some risk of retrospective memory bias [35], and cannot reflect variability within days. In contrast, a recent EMA study reported that the S-allele was positively associated with affective inertia [36], which is argued to reflect emotional *in*flexibility [37]. This finding conflicts with the hypothesis that S-allele induces higher differential susceptibility to environmental influences. However, affective inertia was operationalized as the autocorrelation of emotional states over time. A recent study demonstrated that this indicator might add only limited information to person-means on measures of emotion, leading the authors to suggest the additional assessment of concrete events and contextual information [38].

We used EMA to test whether the *5-HTTLPR* is a differential susceptibility factor for daily events. Participants received questionnaires while they were following their daily routine, repeatedly measuring immediate positive and negative events as well as concurrent positive and negative affect. This allowed us to apply the formal statistical criteria of differential susceptibility [1, 26] and to test genetic differential susceptibility for the first time in the context of ecologically valid experiences. In accordance with the position of Belsky and colleagues [25], we hypothesized homozygous and heterozygous carriers of the S-allele to be more susceptible to events in everyday life than homozygous carriers of the L-allele.

Data and R analysis script are available on the open science framework: https://osf.io/zpvns/?view_only=d1b01be029414512ad835db6192695cd.

## Methods

### Participants

In total, 418 people participated in the EMA study, who were recruited via bulletin boards at the Ruhr University Bochum (Germany). Of these, 326 gave their consent for genetic analyses and were successfully genotyped on the *5-HTTLPR*. To estimate the appropriate sample size for effect detection in EMA studies, prior data with a similar design are necessary [39]. As such data was not available, our sample size was based on the sample size of other genetic EMA studies on the same effect (e.g. [33]).

A simulated post-hoc power analysis based on our empirical data indicated a power of 61.70% to detect a previously reported effect size [33], with a positive predictive value of 92.0% assuming equal prior probabilities. Notably, this previous effect size was smaller than for another study [34] and is therefore more conservative. See the Results subsection on power considerations for further details.

The final sample comprised participants between the ages of 17 and 63 ($M$ = 25.07, $SD$ = 8.62) of which most were female (77%) university students (78%). Other named occupations were full-time work (33%), vocational school (3%), high-school (2%), no occupation (1%), or miscellaneous (1%; some participants endorsed more than one option). The study was approved by the local Ethics Committee of the Faculty of Psychology, Ruhr University Bochum, and all participants provided written informed consent. There was no monetary incentive. University students could receive partial course credit.

### Materials and procedure

#### Genotyping

During an initial laboratory session, saliva samples for DNA extraction were collected in 50 ml tubes via mouthwash before participants were informed about the EMA procedure. DNA was extracted from the mouthwash samples using a salting out procedure [40] with the Master Pure™ DNA Purification Kit. The 43-base pair insertion/ deletion polymorphism was genotyped with a standard PCR reaction as previously described [41]. Ninety-four (29%) individuals were L/L carriers, 152 (47%) were L/S carriers, and 80 (25%) participants were S/S carriers. No deviation from Hardy-Weinberg equilibrium was observed; $\chi^2(1)$ = 1.41, $p$ = .235.

#### EMA

The assessment took place in form of a stratified randomized sampling design over five consecutive days in a first wave ($N$ = 158) and four consecutive days in a second wave ($N$ = 168) to simplify recruitment. Assessments always started on a Thursday to sample both working days and weekend days in a similar frequency. Participants were asked to complete a five-minute survey four times each day, with one random prompt within each of the following fixed intervals: 11.15–12.15 h, 14.00–15.00 h, 17.45–18.45 h, 21.30–22.30 h. During the four intervals, they received an e-mail with a link to the EMA questionnaire and the request to answer the questions promptly. Participants had to possess a smartphone which could notify them when they received the e-mail. They were informed about the importance to honestly fill out as many questionnaires as possible in a timely manner. Compliance was high, with participants returning on average 82% of the questionnaires ($SD$ = 23%, range: 10–100%).

Data preparation was done in IBM SPSS version 23. Nine percent of individual questionnaires were removed because less than 50% of items were completed or time to the preceding questionnaire was below 15 minutes. The latter was necessary as questionnaires were not deactivated after their designated time frame and some participants filled out more than one questionnaire per occasion. These procedures resulted in 4,905 usable questionnaires.

## Measures

The questionnaires included scales on positive and negative affect, as well as positive and negative experiences. Other measures on loneliness and social support were irrelevant for hypotheses concerning the *5-HTTLPR* and are therefore not reported here. Momentary affect was measured by 17 items, of which 13 were selected from the German version of the Positive and Negative Affect Schedule [42]. A two-level exploratory factor analysis for ordered categorical indicators suggested extraction of two factors at both the within- and between-person levels, with acceptable fit indices and clear simple structure after discarding one item (feeling exhausted; see S1 Appendix for details). Reliability estimation for the resulting positive affect (excited; strong; proud; enthusiastic; content; successful; determined; happy) and negative affect scales (distressed; upset; guilty; hostile; irritable; ashamed; nervous; afraid) at the between- and within-person level [43] yielded satisfactory coefficients, indicating high reliability of affect measures: Positive affect $\omega_{within}$ = .88; $\omega_{between}$ = .95; Negative affect $\omega_{within}$ = .80; $\omega_{between}$ = .95.

Positive and negative daily events experienced in the preceding hour were measured with 11 items (seven negative items measuring stressors, four positive items measuring uplifts) regarding different life domains such as personal achievements or interpersonal events (e.g. "I was unsuccessful in my activities", "I had nice moments with other people"). No factor analysis was conducted on these items, as they capture the occurrence of different mostly independent events which are not necessarily correlated. Therefore, estimating a latent (reflective) factor and scale reliability based on item intercorrelations is not meaningful in this case.

Participants responded to all items on a visual analogue scale (range: 0–100) indicating the level of agreement with the statements (upper anchor: "trifft zu" ["applies"]). For each measurement occasion, the items of the four scales (positive affect, negative affect, uplifts, stressors) were averaged into a single momentary scale value, respectively.

## Statistical analyses

The data had a hierarchical structure with measurements (level 1) nested within days (level 2), nested within participants (level 3). A likelihood ratio test for an intercept-only model with mood as the dependent variable indicated that including the day level increased model fit significantly ($\chi^2(1)$ = 55.34, $p < .001$; 46.6% variance between participants, 11.7% between days, and 47.7% between measurement occasions). Multilevel analyses were conducted in R (version 3.5.1) using the package *nlme* to accommodate the dependencies in the data, with a first order continuous time autoregressive (CAR1) covariance structure on level 1. Level-1 predictors were centered on the person mean, so they represent pure within-person effects [44]. The person means of these predictors were centered on the grand mean and entered as level-3 predictors (see S1 Appendix for model equations). Even though not integral to our main hypothesis, person means were included for the comparison with past cross-sectional studies and for usage in future meta-analyses, as they are statistically orthogonal to the within-person predictor and therefore do not interfere with the within-person effects.

The trichotomous genotype was recoded into two Helmert contrasts in concordance with Gunthert et al. [33]: One contrast represents the mean difference between the L/L and the

pooled S-carriers, with a positive sign indicating a larger association for the S-carriers. This contrast tests for the most common assumption that the S-allele is dominant. The other contrast represents the mean difference between the L/S- and the S/S-carriers, with a positive sign indicating a larger association for the S/S-carriers. This contrast tests for the additional possibility that the effect of the S-allele is additive (see S1 Appendix for details). In many previous studies, the comparison between the L/L-carriers and the carriers of at least one S-allele has been the central contrast of interest, while the contrast between L/S and S/S carriers is often not reported [27].

First, all analyses were conducted on the full sample of 326 participants. Then, to test the robustness of the results, all models were reanalyzed after applying stronger criteria for careless responding, excluding participants who returned less than 33% of questionnaires or either had a negative Cronbach's alpha or an intra-individual standard deviation below 1 on positive or negative affect [45]. Following these criteria, 47 participants (14.4%) were excluded. The main results remained unaltered by these data exclusions. Therefore, analyses on the full sample are reported.

## Results

### 5-HTTLPR as differential susceptibility factor

To confirm that a trait is a differential susceptibility factor, several criteria must be met [1, 26]. First, predictor and outcome should be on continuous scales comprising negative and positive aspects. Therefore, for each measurement, average stressors were subtracted from average uplifts to create a single index of environment valence (predictor); average negative affect was subtracted from average positive affect to create a single index of mood valence (outcome). Both variables ranged between −100 and 100 with higher values indicating that positive aspects outweigh negative aspects. Descriptive statistics for these measures and their components can be found in Table 1.

Second, the susceptibility factor must not be correlated with either the predictor or the outcome. To test this, null models without predictors were compared to models including only the two genotype variables as level-3 predictors with a likelihood ratio test. Genotype had no main effect on the environment index, $\chi^2(2) = 0.01$, $p = .997$, nor on the mood index, $\chi^2(2) = 0.38$, $p = .827$, fulfilling this criterion and providing no evidence for gene-environment correlation.

The most important criterion for differential susceptibility is that there must be a crossover interaction between the susceptibility factor and the predictor, in our case the *5-HTTLPR* and the environment index. On the within-person level, the *5-HTTLPR* moderated the effect

**Table 1. Descriptive statistics.**

| Measure | *Mean* | $SD_{between}$ | $SD_{within}$ | *Range* |
|---|---|---|---|---|
| Mood | 32.83 | 19.54 | 22.24 | −86–100 |
| Positive Affect | 43.83 | 15.38 | 16.71 | 0–100 |
| Negative Affect | 11.00 | 8.81 | 9.93 | 0–90 |
| Environment | 19.52 | 14.88 | 26.37 | −85–100 |
| Uplifts | 40.33 | 13.28 | 21.49 | 0–100 |
| Stressors | 20.82 | 7.66 | 14.49 | 0–100 |

Table depicts weighted means, standard deviations between and within persons as well as the empirical range of the continuous scales.

$N$ = 326; total number of observations = 4,905.

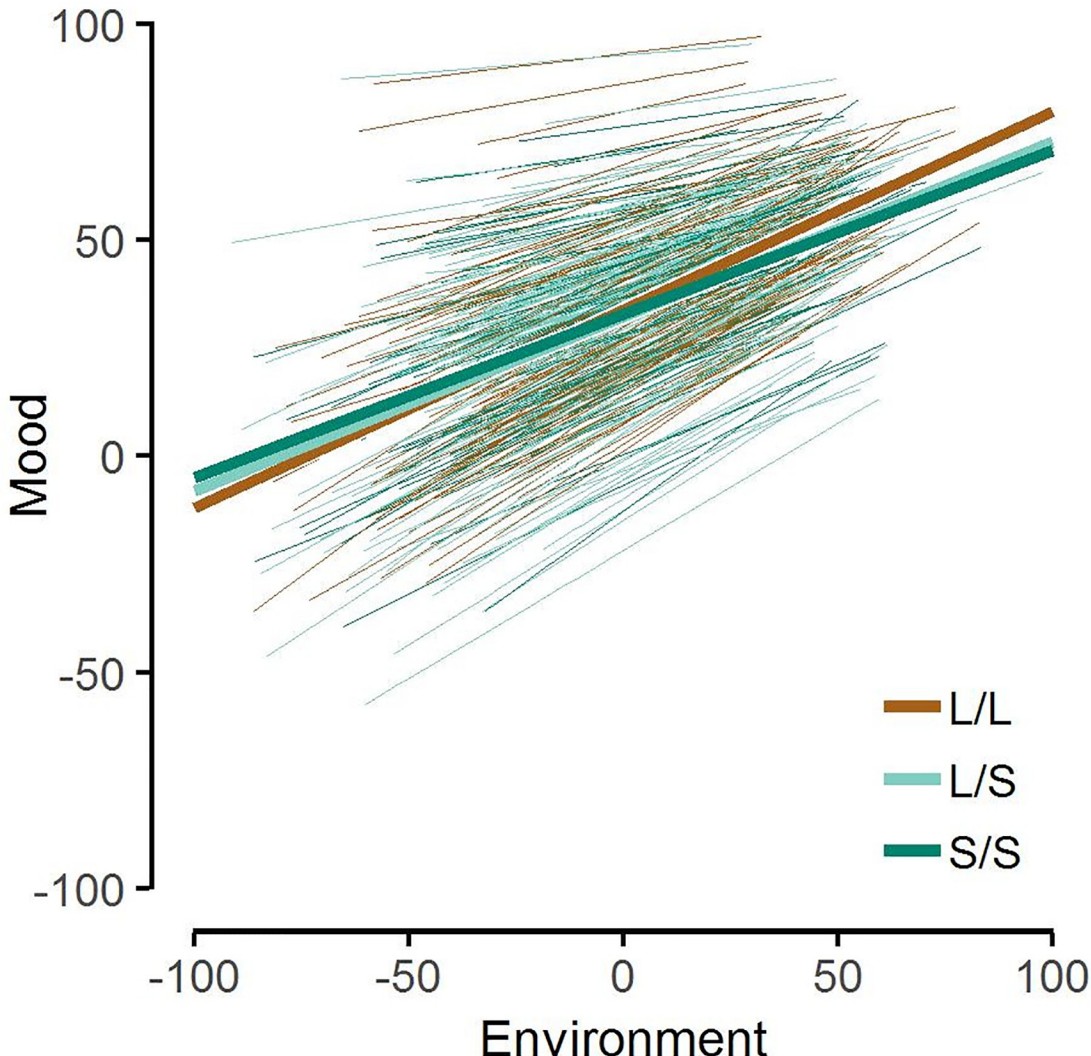

**Fig 1. Mood predicted by the cross-over interaction between environment and genotype.** Thick lines represent average environmental reactivity for different genotype groups. Thin lines represent environmental reactivity for individual participants.

of environmental influences on mood, with L/L-carriers reporting more positive mood in positive environments and more negative mood in negative environments (Fig 1). Genotype explained 1.69% of the variance in reactivity slopes. Compared to S-carriers, L/L-carriers had a larger environmental reactivity ($\gamma_{101} = -0.07$, *SE* = 0.03, 95% CI = [−0.13, −0.00], *p* = .036), with no difference between the two S-carrying groups ($\gamma_{102} = -0.02$, *SE* = 0.04, 95% CI = [−0.09, 0.05], *p* = .560).

As afforded by differential susceptibility, the interaction had a cross–over shape, meaning the intersections of the genotype lines were around the middle of the predictor scale. Results of a bootstrapping procedure (see S1 Appendix for details) showed that the cross-over point of the regression lines of the L/L- and the pooled S-groups as well as its confidence intervals were within the limits of the scale, *M* = −14.88, 95% CI = [−74.05, 25.90], fulfilling the last differential susceptibility criterion [26].

On the between-person level, the *5-HTTLPR* did not reliably moderate the relationship between mood and the average environment of a person. This was the case for both the

contrast between L/L- and pooled S-carriers ($\gamma_{004}$ = 0.05, *SE* = 0.10, 95% CI = [−0.15, 0.24], *p* = .634) as well as between L/S- and S/S-carriers ($\gamma_{005}$ = −0.15, *SE* = 0.12, 95% CI = [−0.39, 0.09], *p* = .210).

In sum, the *5-HTTLPR* met all statistical criteria for differential susceptibility on a within-person level. Participants homozygous for the L-allele were more reactive to environmental influences than S-allele carriers.

## Separating effects of stressors and uplifts

The analysis described above represents a formal statistical test of susceptibility which ideally requires a continuum from positive to negative [1], realized here with difference scores. We found a global pattern of enhanced reactivity in L/L-carriers. Next, we tested whether this effect could be attributed to a higher reactivity to uplifts, stressors, or both. We therefore computed four separate models, with either positive or negative affect as the outcome and either stressors or uplifts as the predictor.

**Positive affect.** Overall, uplifts were associated with higher positive affect on the within-person level ($\gamma_{100}$ = 0.34, *SE* = 0.01, 95% CI = [0.31, 0.36], *p* < .001) and the between person-level ($\gamma_{001}$ = 0.75, *SE* = 0.04, 95% CI = [0.66, 0.84], *p* < .001). Analogously, stressors were associated with *lower* positive affect on the within-person level ($\gamma_{100}$ = −0.22, *SE* = 0.02, 95% CI = [−0.26, −0.18], *p* < .001) and the between-person level ($\gamma_{001}$ = −0.26, *SE* = 0.11, 95% CI = [−0.48, −0.04], *p* = .019). Hence, positive affect was associated with both positive and negative experiences.

The *5-HTTLPR* moderated the within-person association between positive affect and both stressors and uplifts (Table 2). The positive affect of L/L carriers was significantly more positively related to uplifts and more negatively related to stressors compared to S-carriers (Fig 2). There was no further reliable difference between L/S- and S/S-carriers in the association between positive affect and stressors or uplifts (Table 2). *5-HTTLPR* genotype explained 5.76% and 4.73% of variance in reactivity slopes, respectively. In sum, these results corroborate that the *5-HTTLPR* is a differential susceptibility factor, with L/L-carriers being more susceptible to environmental influences than carriers of the S-allele.

As in the main analysis, there were no reliable gene–environment interactions on the between-person level (S1 Table).

**Negative affect.** Complementary to the results for positive affect, stressors were associated with higher negative affect on the within-person level ($\gamma_{100}$ = 0.27, *SE* = 0.02, 95% CI = [0.24,

**Table 2. Gene–environment interactions predicting affect on the within-person level.**

| Predictor | L/L vs S | | | L/S vs S/S | | |
| --- | --- | --- | --- | --- | --- | --- |
| | Estimate (*SE*) | 95% CI | *p* | Estimate (*SE*) | 95% CI | *p* |
| *Positive Affect* | | | | | | |
| Uplifts | −0.09 (0.03) | [−0.15, −0.03] | .002 | 0.01 (0.03) | [−0.06, 0.07] | .836 |
| Stressors | 0.11 (0.05) | [0.02, 0.20] | .020 | 0.07 (0.05) | [−0.03, 0.17] | .195 |
| *Negative Affect* | | | | | | |
| Uplifts | 0.00 (0.02) | [−0.03, 0.04] | .782 | 0.02 (0.02) | [−0.02, 0.05] | .415 |
| Stressors | 0.05 (0.03) | [−0.02, 0.11] | .138 | −0.05 (0.04) | [−0.12, 0.03] | .210 |

The first three result columns report the contrast of the L/L- against the pooled L/S- and S/S-carriers. A positive sign indicates a more positive slope in the S-groups. The three rightmost columns report the contrast of the L/S against the S/S carriers. Here, a positive sign indicates a more positive slope in the S/S-group, compared to the L/S-group.

*N* = 326; number of person days = 1,483; total number of observations = 4,905.

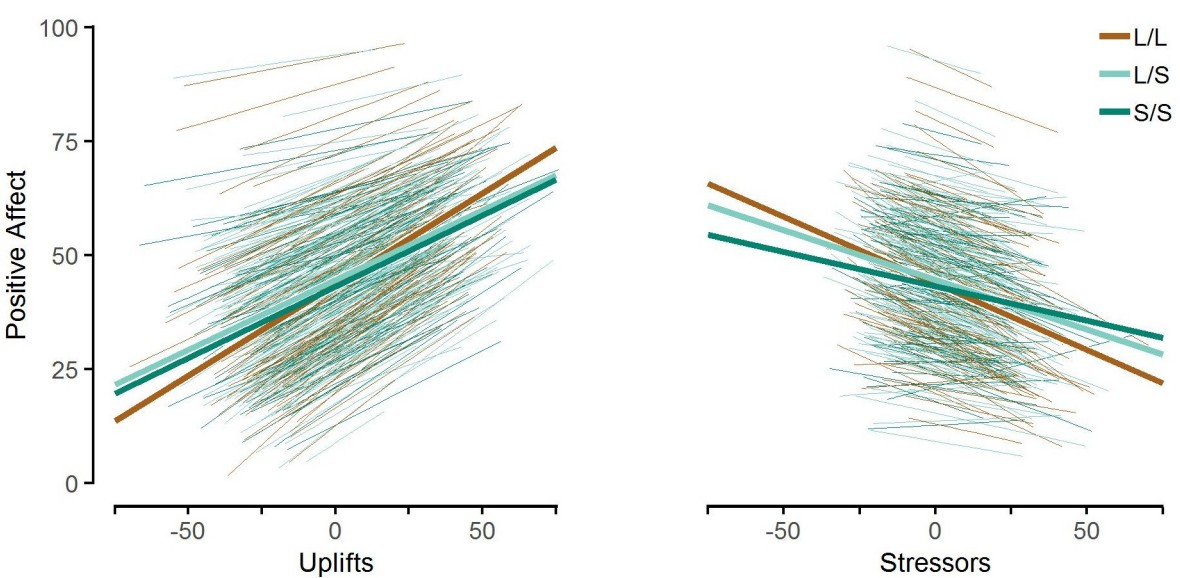

**Fig 2. Positive affect predicted by uplifts, stressors, and their interaction with genotype.** Thick lines represent average environmental reactivity for different genotype groups. Thin lines represent environmental reactivity for individual participants.

0.30], $p < .001$) as well as the between-person level ($\gamma_{001} = 0.61$, *SE* = 0.04, 95% CI = [0.52, 0.70], $p < .001$). The association between uplifts and negative affect on the within-person level was weaker, but still significantly different from zero ($\gamma_{100} = -0.06$, *SE* = 0.01, 95% CI = [−0.07, −0.04], $p < .001$). On the between-person level, this effect was not significant ($\gamma_{001} = -0.06$, *SE* = 0.03, 95% CI = [−0.13, 0.01], $p = .106$). To summarize, negative affect appeared to be mainly associated with negative rather than positive events.

The *5-HTTLPR* moderated none of the relationships between experiences and negative affect, neither on the within-person level (Table 1) nor on the between-person level (S2 Table). Diagnostic plots revealed marked non-normality in residuals, likely due to the inherent positive skew in negative affect. Therefore, a multilevel model for ordered categorical outcomes was conducted on the relationship between negative affect and stressors (S1 Appendix, S2 & S3 Tables). This analysis yielded the same results: There was no evidence for the *5-HTTLPR* moderating the association of stressors and negative affect at any level of analysis.

## Exploratory analyses on the effect of time resolution

In contrast to our findings, past EMA studies pointed towards enhanced stress reactivity of negative emotions and psychopathological symptoms in carriers of the S-allele [33,34]. These studies differ from ours in that they employed only one daily measurement which retrospectively assessed stressors for the whole day. The time gap between event and affect measurement is therefore likely to be much larger than the one-hour restriction we imposed for events to be reported. Such differences in time resolution can crucially affect the results of EMA studies [46]. Nevertheless, both approaches might capture different relevant aspects of affective functioning. For example, while a higher time resolution might be more likely to capture the degree of reactivity (e.g. when affect is measured closer to the response peak after an event), a lower time resolution might be more likely to capture the speed of recovery and more enduring affective consequences.

We approximated the lower time resolution of prior studies [33,34] by regressing only the last daily negative affect measure on average daily stressors of the same day, because those

studies measured affect at the end of the day and retrospective stressors over the course of the whole day. This reduced our statistical model to two-levels with days nested within individuals. We chose the regression of negative affect on stressors, because it most closely resembles the constructs used in the two EMA studies.

Descriptively, the S/S group exhibited the strongest and the L/L group the weakest contingency between stressors and negative affect, with the L/S group lying in between (S1 Fig), as was reported in previous studies. To accommodate the skewness of negative affect, we again conducted an ordinal regression (S4 Table), where the difference between the S/S and the L/S group came close to reaching statistical significance ($OR$ = 1.05, 95% CI = [1.00, 1.10], $p$ = .053). These inferential tests should be interpreted with caution, as they were exploratory and the inclusion of only one daily measurement made them statistically underpowered. Still, they highlight the possibility of effect reversals when stress reactivity is measured on different time scales.

### Power considerations

The replicability of candidate gene studies is increasingly criticized as effect sizes are likely small, leading to low statistical power and, consequently, a low probability for significant effects in the literature to be true positives [20,47]. Recently, flexible simulation-based power analyses for multilevel models have become more accessible [48]. Therefore, we tested the statistical power of our design based on our empirical covariance structure and the effect sizes reported by Gunthert et al. [33] who found a moderation of the stress–anxiety relationship by the *5-HTTLPR* and replicated this effect on the same sample one year later. We used the study by Gunthert et al. [33] as a prior, as it resembles our study most closely in terms of measures and design, even though they had a lower sampling frequency.

We followed the procedure described in Green and Macleod [49]. First, we refitted our main model with the lmer function of the package lme4 (a necessity for the procedure with the disadvantage of not being able to model the AR1 covariance structure used in the nlme model [50]). Then, we substituted our empirical gene–environment interaction with the effect size from Gunthert et al. [33], averaged over the two replications (r = .06). Last, we conducted the simulation-based power analysis with the simr package [49].

The simulation indicated that we had a power of 61.70% to detect the effect size reported by Gunthert et al. [33]. This number can be used to estimate the positive predictive value, the probability that a significant finding is a true positive [47]. Assuming that H0 and H1 are equally likely, the probability that a significant finding in our design is true is .93.

Both the true effect size and the prior probability of H1 and H0 are unknown. Therefore, we also calculated the positive predictive value for more pessimistic scenarios. Assuming that the H0 is two times more likely than the H1, the probability that a significant finding is a true is .86. When the statistical power is reduced to 50%, accounting for the possibility of inflated effect sizes in Gunthert et al. [33](2007), this value drops only three percent points to .83. With equal prior probabilities for H1 and H0 and a power of 50% the positive predictive value is .91.

### Discussion

People differ in the way they respond to environmental challenges and opportunities, and part of this variability is explained by genetic variation [7]. Here, we tested whether the *5-HTTLPR* is associated with differential affective responses to positive and negative daily events in a 'for better and for worse' manner [1]. We found that the *5-HTTLPR* met all formal statistical criteria of a differential susceptibility factor. Contrary to our expectation, not the S-allele carriers, but the L/L carriers exhibited a stronger contingency between environmental influences and

affect. Follow-up analyses revealed that particularly their positive affect was more strongly associated with both positive and negative events.

As the S-allele has been generally regarded as the more plastic allele [27,28], the finding that L/L-carriers were the more susceptible group in our study might seem contradictory. However, this is not necessarily the case. In contrast to most studies on the *5-HTTLPR*, the main purpose of our study was to investigate environmental reactivity on a within-person level. Expecting within- and between-person analyses must yield the same results constitutes a so-called ecological fallacy [51] and the necessary assumptions which equate the two levels are rarely met [29]. Nevertheless, all meta-analyses which do report reliable effects on negative outcomes like PTSD [52] did in fact find the S-allele to be a risk-allele. Our findings do not contradict these meta-analytic findings, because higher reactivity of positive affect to typical daily events, as observed in L/L carriers in our study, does not necessarily imply someone is more prone to develop pathological symptoms.

Emotions have an adaptive function as they allow us to respond flexibly to changing external demands [53]. Therefore, to a certain degree, a higher environment-affect contingency likely reflects a desirable trait. A recent EMA study showed that S-carriers have a higher affective inertia, meaning their affective state is more likely to carry over from one moment to the next [36]. This kind of affective inflexibility is related to psychopathology and generally lowered psychological well-being [37,54]. Specifically, depression is often accompanied by context insensitivity to both positive and negative stimuli [55], the pattern we found in our S-carrying groups. Hence, the S-allele might be a genetic vulnerability factor that contributes to emotional inflexibility [36], which in turn prospectively predicts psychopathology [56]. Still, more research is needed to support this possible explanation as our study design is not sufficient to link environmental reactivity with adaptive emotional functioning.

Importantly, our finding precludes neither that carriers of the S-allele possess neural networks that are more excitable on a scale of seconds [23], nor that they are more susceptible to influences in earlier, developmentally sensitive periods [27]. On the contrary, our exploratory analyses demonstrated that investigating reactivity on different time scales can produce opposing patterns. Therefore, connecting reactivity to naturalistic events in adults, as investigated here, with neurobiological nanoscale and developmental macroscale data can lead to a more nuanced perspective on the phenotypic expression of reactivity-related genes.

## False positives in candidate gene studies

Candidate gene studies are increasingly facing scrutiny, and this is especially true for the *5-HTTLPR*. Robust meta-analyses have been published showing that there is no moderating effect of the *5-HTTLPR* in the association between (early) life stress and depression [19,20]. This lack of moderation with regard to a very heterogeneous disorder entity, measured as a dichotomous outcome, has led many researchers to dismiss all *5-HTTLPR* findings as spurious, including those studying more refined phenotypes. Nevertheless, effect sizes for candidate genes vary according to the phenotype measured. Specifically, for the *5-HTTLPR* it has been shown that effect sizes are smaller for categorical clinical diagnoses than for dimensional measures, while effect sizes for neurobiological outcomes are largest [57]. Within-person models of momentary experiences likely lie in between the last two categories [32]. Hence, nonexistent or small genetic associations with brief questionnaires categorically measuring complex mental disorders which exhibit strong within-group heterogeneity and between-group comorbidity are not necessarily a good prior for effect sizes in studies which focus on more basic functions like the general sensitivity to experiences.

We conducted a power analysis based on two prior reported effect sizes from a study similar to ours in terms of both hypothesis and design [33]. We found that although our study is likely not optimally powered, the probability that a significant finding is a false positive in our design is low, even in scenarios were true power would be lower as our empirical estimate and prior odds in favor of the H0.

Moreover, our hypothesis is based on significant meta-analyses on the *5-HTTLPR* and affective functioning [23,24], a developmental meta-analysis specifically implicating the *5-HTTLPR* in differential susceptibility [27] and several EMA studies on affective processes [33,34,36]. Even though publication bias tended to be present in the meta-analyses, they did not nullify their effect sizes, which still remained larger than those employed in our power-analysis. Even in the absence of respective genome-wide association studies (GWAS), these meta-analyses should not be readily cast aside in the argument whether research on classical candidate genes should be further pursued. Until GWAS become available which feature deep phenotyping in samples testing hundreds of thousands of participants, studies with plausible candidates tailored to specific phenotypes might still have their merit.

## Limitations

Our study has several limitations. First, we did not find the same pattern for negative affect, which might be due to its undesirable statistical properties. Negative affect is prone to floor effects, which leads to a skewed distribution and low variance. Still, nonlinear models did not yield any significant effects on negative affect as well. Alternatively, some theorists proposed that positive and negative affect have different neural substrates [58], which might also be differentially influenced by the *5-HTTLPR*. However, a meta-analysis on almost 400 neuroimaging studies concluded that affect is most likely represented in complex activity patterns of valence general neuron populations instead of two separate systems [59]. How genes affect these patterns remains an interesting target for future research. Also, it might be of interest whether the flexibility of positive affect has more adaptive value than flexibility of negative affect. The meta-analysis by Houben et al. [37] found weaker relationships between well-being and positive affective stability measures than for negative affective stability. Nevertheless, studies measuring the contingency to external events where not included in the meta-analysis and context-dependent reactivity might differ from context-free variability as a marker of affective flexibility. Second, about 80% of our sample were German university students, making the generalizability to other populations uncertain. Third, although our study targets the process of environmental reactivity, which assumes a causal contingency between stimulus and output, our design is still correlational in nature. It is for example possible that momentary affect contaminates the ratings of previous stressors.

Fourth, we combined the two environment variables and the two affect variables in one index, respectively, by calculating difference scores. This assumes that a positive environment is one where positive events outweigh negative events and that a positive state is one where positive affect outweighs negative affect. Therefore, the main analysis cannot account for mixed environments or mixed states [60], where both positive and negative aspects are strongly prevalent at the same time. Still, so far no study on differential susceptibility accounted for mixed environments or outcomes and this procedure allowed us to apply formal statistical methods to attest differential susceptibility with the most widely used scales.

Fifth, our operationalization of environmental events through a semi-continuous visual analogue scales raises the question whether these measures are confounded by momentary affect. Still, the alternative of using binary indicators for event occurrence has shortcomings as well. For example, a person could indicate interpersonal conflicts due to very minor or

extremely severe events, which has vastly different implications. Consequently, high stress reactivity might reflect more severe events. Also, binary indicators might not be sufficiently sensitive to capture minor stressors, which we specifically aimed for in our study.

Sixth, we did not genotype for rs25531 A/G SNP within the 5-HTTLPR repetitive element, as suggested during the review process, because a reanalysis of the available DNA samples was not possible due to covid-19-associated lab closure. The SNP has been reported to render the G allele within the l allele transcriptionally less efficient [61], a finding that has not been consistently replicated [62].

Last, although the effect sizes were reasonable for genetic studies on single polymorphisms, they are not large in absolute terms. Nevertheless, the main purpose of the present study was to test the theory that individual variability in the serotonin system is related to differential susceptibility. Searching for reliable effects of serotonergic genes is likely currently the only viable approach to test this theory in humans in daily life even if these effects are small.

## Conclusion

We found that the *5-HTTLPR* is a differential susceptibility factor for affective reactivity to daily events. This finding confirms the serotonin system's proposed role in general reactivity to both positive and negative environments [13]. Contrary to the general (even though still controversial) notion of most macro longitudinal studies, instead of S-allele carriers, L/L-allele carriers were the more susceptible group, which might reflect adaptive emotional flexibility. To come to a more complete view of genetic influences on susceptibility, we have to combine macro- and microscale approaches, synthesizing major life experiences (e.g. early adversity) and developmental trajectories of environmental reactivity [2]. For this, we need reliable, ecologically valid markers to characterize a person's individual reactivity at one point in life. EMA is a promising candidate for this task.

## Supporting information

**S1 Appendix. Supplemental methods.**
(DOCX)

**S1 Fig. Effect of average daily stressors on the last daily measure of negative affect by 5-HTTLPR genotype.**
(JPG)

**S1 Table. Gene–environment interactions predicting affect on the between-person level.**
(DOCX)

**S2 Table. Ordinal mixed regression of negative affect on stressors.**
(DOCX)

**S3 Table. Ordinal mixed regression of negative affect on uplifts.**
(DOCX)

**S4 Table. Ordinal mixed regression of last daily negative affect measure on average daily stressors.**
(DOCX)

## Author Contributions

**Conceptualization:** Wolff Schlotz, Robert Kumsta.

**Formal analysis:** Maurizio Sicorello, Dirk Moser, Wolff Schlotz.

**Funding acquisition:** Robert Kumsta.

**Investigation:** Linda Dieckmann.

**Methodology:** Maurizio Sicorello, Dirk Moser, Andreas B. Neubauer, Wolff Schlotz.

**Project administration:** Robert Kumsta.

**Resources:** Robert Kumsta.

**Supervision:** Andreas B. Neubauer, Robert Kumsta.

**Visualization:** Maurizio Sicorello.

**Writing – original draft:** Maurizio Sicorello, Robert Kumsta.

**Writing – review & editing:** Linda Dieckmann, Vanessa Lux, Maike Luhmann, Andreas B. Neubauer, Wolff Schlotz, Robert Kumsta.

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
