## [Decision Letter · Decision Letter 0]

17 Mar 2020

PONE-D-20-03284

Highs and lows: Genetic susceptibility to daily events

PLOS ONE

Dear Mr. Sicorello,

Thank you for submitting your manuscript to PLOS ONE. After careful consideration, we feel that it has merit but does not fully meet PLOS ONE’s publication criteria as it currently stands. Therefore, we invite you to submit a revised version of the manuscript that addresses the points raised during the review process.

This is a very interesting study and an overall well written manuscript. I think the reviewers make some relevant suggestions and comments for a revision, which I encourage you to follow for the most parts.

We would appreciate receiving your revised manuscript by May 01 2020 11:59PM. To enhance the reproducibility of your results, we recommend that if applicable you deposit your laboratory protocols in protocols.io, where a protocol can be assigned its own identifier (DOI) such that it can be cited independently in the future. For instructions see: http://journals.plos.org/plosone/s/submission-guidelines#loc-laboratory-protocols

We look forward to receiving your revised manuscript.

Kind regards,

Hedwig Eisenbarth

Academic Editor

PLOS ONE

Journal Requirements:

Additional Editor Comments (if provided):

Aii suggest that you follow both reviewer's suggestions which mainly contain several points of clarification.

Reviewers' comments:

Reviewer's Responses to Questions

**Comments to the Author**

1. Is the manuscript technically sound, and do the data support the conclusions?

Reviewer #1: Yes

Reviewer #2: Yes

2. Has the statistical analysis been performed appropriately and rigorously? 

Reviewer #1: Yes

Reviewer #2: Yes

3. Have the authors made all data underlying the findings in their manuscript fully available?

Reviewer #1: Yes

Reviewer #2: Yes

4. Is the manuscript presented in an intelligible fashion and written in standard English?

Reviewer #1: Yes

Reviewer #2: No

5. Review Comments to the Author

Reviewer #1: PONE-D-20-03284

Highs and lows: Genetic susceptibility to daily events

Sicortello et al. tested whether genetic variation in the serotonin transporter (5-HTTLPR) is a differential susceptibility factor for daily events. Participants completed smartphone

questionnaires four times a day over four to five days, measuring stressors, uplifts,

positive and negative affect. Results showed that positive affect in carriers of the short allele (S) were less reactive to both uplifts and stressors. This pattern might reflect relative affective inflexibility in S allele carriers.

• The study is well written, the topic of high interest and the statistical analysis using an EMA approach is creative.

• However, the duration of data collection 4-5 days is rather short. The possibility to be confronted with a major stressor is rather low and if it happens this might be a spurious effect.

• There are no clear hypotheses. Contrary to the results one would expect higher mood variability in s-allele carriers with respect to negative emotions.

• The authors argue that 5-HTTLPR is of interest due to its associations to personality traits of negative emotionality. Have the authors assessed measures of personality? Normally one would expect that personality explains far more variance than a single genetic polymorphism. Explained variance < 2 %.

• The authors should in addition genotype for rs25531 which would allow a triallelic approach in evaluating their data. This is nowadays common standard in the analysis 5-HTTLPR. (Hu X, Oroszi G, Chun J, Smith TL, Goldman D, Schuckit MA . An expanded evaluation of the relationship of four alleles to the level of response to alcohol and the alcoholism risk. Alcohol Clin Exp Res 2005; 29: 8–16.)

Reviewer #2: Review of PONE_D_20-03284

The study reported in this manuscript examines whether 5-HTTLPR operates as a genetic differential susceptibility factor for affective responding to daily events. The study addresses a highly relevant research question in a way that is novel in several aspects. It is based on interesting data from a reasonably sized sample, and uses sophisticated data-analytic techniques to tests its predictions. Nevertheless, I identified a few issues that deserve additional attention. I explain these below, along with some impressions and observations (which are a matter of taste rather than a formal evaluation of the paper).

Abstract: I think the abstract should included some information on the sample and its size, (and maybe also on the analytic approach taken). Also, I suspect that the last sentence does not correspond to what the authors wanted to express (which is probably that the study provides insight into the serotonin system’s general role…).

General point: In the sentence “Positive affect in carriers of the short allele…”, as well as in other parts of the manuscript, the authors make a comparison, but name only one comparison group (in this sentence the S-allele carriers), but not to which they are compared. I think these statements would gain clarity from complete descriptions of the comparisons.

P3, last sentence (minor): I think that the reference to the compatibility of the DS with the heritability patterns from twin studies needs a bit more explanation to fulfill its function here.

P4, 75ff. (minor): This second part of the paragraph is quite cursory and therefore not very informative and not very helpful.

P4, 81ff.: When the 5-HTTLPR is introduced, it would be interesting if in a sentence or two, the authors would say something about the function of the region.

P5, first paragraph: I think this paragraph needs some revision. The first observation the authors name does not support the hypothesis for the 5-HTTLPR to be a differential susceptibility factor. This first observation simply points to gap in research. It is not clear to me what the authors want to point to with the Sagy & Antonovsky 2000 reference. Then the second observation provides some ground for a DS hypothesis, but it is not the fact that most studies did not further examine this, that is informative. Finally, the “consequently” doesn’t really fit here to introduce the first partial sentence, and similarly, the second partial sentence doesn’t follow logically after the “but”.

Also related to this paragraph, I would find it important if the authors provided some insight into the literature and maybe a few sample studies that point to susceptibility of positive experience (e.g., Koenen’s work or Way at al., 2006).

P5, 114ff.: I think this paragraph addresses a very important point and strength of this study. However, the last sentence, as formulated, may be correct but is not specific to the DS hypothesis, as it would apply as well to a vulnerability hypothesis.

General point about EMA data and reactivity: This part has an implicit causal “touch”, and so it should be noted somewhere that we cannot draw conclusions about causality or a causal direction, since these are correlational data.

P6, 2nd paragraph, 130ff: The reference to the van Roekel et al study is interesting, but I think more detail is necessary to make this point with sufficient clarity. There is a long way from within-subject associations of daily experiences and affect, to AR(1)measures of emotional inertia, to the concept of emotional flexibility. The second sentence, at 132 ff., is unclear and maybe incomplete.

Section “Participants”: This section suggests that an a priori power analysis has been conducted to determine the necessary sample size. Therefore, the authors should report the sample size determined to achieve adequate power here. They can still give the details on the power analysis later on.

P8, 193ff (minor): Because the authors begin a new paragraph, it was not immediately clear to me that the EFA still referred to the PANAS.

Daily events: Could the authors provide more detail on this measure? It would be useful to know the instruction, and specially the labels used with the visual analogue scale. I was a bit surprised to see that the occurrence of an event was measured on a 0-100 scale, given that the scale measures events (but as someone who does a lot of EMA research I see the advantage of this approach). My concern would be that this kind of measures also taps into mood, and might spuriously inflate correlations with panas measures.

I was glad to see that the authors didn’t report an alpha for the daily events measure, and fully agree with their justification! Something that I was wondering about was whether this wouldn’t apply to the PANAS, too. Probably not when it is used to measure mood-like affective states without particular emotional experiences occurring. Given the consistency scores reported, this seems to be the case here. This might be relevant to the interpretation of the results, since we’re probably not dealing with distinct emotional responses here, but rather with moderate shifts in participants mood.

P10, 224: Could the authors say more about the lack of power they refer here?

P10 230: I think it would be useful if the authors provided their rationale for using these contrasts here. Why not contrasting both groups of S allele carriers from the homozygous L allele carriers?

P11: Was there an association between genotype and variability on affect and daily experience variables?

P11, 262, minor: “Compared to” might fit better here than “instead”.

P13, 299-305 (minor): This paragraph is not very precise and therefore not easy to read (beginning: do you refer to one or more associations?). No difference on what? … more susceptible to what? From the context, one can figure out what you mean, but complete and more precise formulations would help.

Results, general point: The effects, particularly that of testing differential susceptibility, is rather small. Did the authors expect an effect of this size, and do they consider it relevant nevertheless? I think it would be important that the authors commented on the effect sizes.

Discussion:

General point: The discussion is relatively brief, if the piece on false positives is not counted. I would have hoped for a more elaborate discussion of the specifics of what the authors have actually studied.

The study period was quite brief, and similar studies collecting binary indications of daily events have occurred find relatively few such events, particularly negative ones, with a sizable portion of the sample report no negative events at all within a 4- or 5-day period. Are the authors convinced that all participants experienced clearly positive and negative events during the 4 or 5 days? This may point to the possibility that what was studied here was not necessarily emotional reactivity to events, in the sense of significant responses with an adaptive goal, but rather mild fluctuations of positivity as a function of relatively ordinary experiences. Some situations may be evaluated as more or less positive or negative, without representing a challenge or opportunity that would deserve a specific response. The average of uplifts and stressors seem clearly above 0, but given that participants probably reported with a finger on a smartphone display (not sure that this is how it was done), it may still be that much of the ratings at the lower 10-20% of the scale range may represent indications of no major event occurring. At least this is the experience we’re making with these kinds of tools.

All these possibilities may or may not have led to an attenuation of the “real” moderator effects. It could be interesting to see whether setting the threshold for registering an event to a value above the lowest 25% of the VAS would yield the same results.

P18, 390 (minor): Although the authors note that they expect stronger associations for s allele carriers in the intro, it was not too clear to me that they did have this expectation since it is not marked clearly as a hypothesis.

P19, top: the reference to the dynamic system is unclear. I suggest that the authors either provide more detail about it, or delete the reference. In the next sentence, it is not clear what levels the authors refer to. Generally, I think this sentence must be rewritten. I think I know what they authors mean, but the sentence lacks precision and sufficient information for the reader less familiar with the topic.

General remark about the discussion: I find it important and interesting that the authors try to connect these results with other phenomena related to emotional adaptation and adjustment, and patterns of emotion patterns. However, this research did not examine flexibility or emotion patterns, and sometimes the authors risk to be overly speculative. Also, the cover a large number of different topics, each one hinting at some research and touching those only very superficially. Several parts of the discussion are therefore accessible only to the readers deeply familiar with the topic. Less (with more detail) would be more here.

The following papers may be relevant to this study.

Schoebi, D., Way, B. M., Karney, B. R., & Bradbury, T. N. (2012). Genetic moderation of sensitivity to positive and negative affect in marriage. Emotion, 12(2), 208.

Haase, C. M., Saslow, L. R., Bloch, L., Saturn, S. R., Casey, J. J., Seider, B. H., ... & Levenson, R. W. (2013). The 5-HTTLPR polymorphism in the serotonin transporter gene moderates the association between emotional behavior and changes in marital satisfaction over time. Emotion, 13(6), 1068.

6. PLOS authors have the option to publish the peer review history of their article (what does this mean?). If published, this will include your full peer review and any attached files.

Reviewer #1: No

Reviewer #2: Yes: Dominik Schoebi

---

## [Author Response · Author response to Decision Letter 0]

15 Apr 2020

PONE-D-20-03284

Highs and lows: Genetic susceptibility to daily events

Response to Reviewers

We first like the thank the two reviewers as well the editor for their contributions. Below, you will our responses to the specific remarks.

Reviewer #1:

Comment 1: The study is well written, the topic of high interest and the statistical analysis using an EMA approach is creative.

Response: We thank the reviewer for the positive overall evaluation. 

Comment 2: The duration of data collection 4-5 days is rather short. The possibility to be confronted with a major stressor is rather low and if it happens this might be a spurious effect.

Response: We completely agree with the reviewer that encountering major stressors is not sufficiently likely in this design. The main purpose of our study was to focus on minor stressors - or daily hassles - which we tried to express in the first sentence and the following sentence of the introduction, but with insufficient clarity. We added the italicized part to the following sentence to highlight this point:

“EMA methods are capable of measuring even minor daily experiences which often have a higher predictive utility than major life events and were the main target in the present study (DeLongis, Coyne, Dakof, Folkman, & Lazarus, 1982; Vinkers et al., 2014)”

The goal to capture minor daily stressors was also the reason for our continuous operationalization of stressors. We added a discussion of this to the limitation section, including the following argument:

“Also, binary indicators might not be sufficiently sensitive to capture minor stressors, which we specifically aimed for in our study.”

Comment 3: There are no clear hypotheses. Contrary to the results one would expect higher mood variability in s-allele carriers with respect to negative emotions.

Response: Our hypothesis was stated at the end of the introduction: 

“In accordance with the position of Belsky and colleagues (Belsky et al., 2009), we expected homozygous and heterozygous carriers of the S-allele to be more susceptible to events in everyday life than homozygous carriers of the L-allele”.

We exchanged the word “expected” for “hypothesized” to make this clearer.

In agreement with the reviewer’s comment, the first paragraph of the discussion and the conclusion subsection had included the following statements:

“Contrary to our expectation, not the S-allele carriers, but the L/L carriers exhibited a stronger contingency between environmental influences and affect”

“Contrary to the general (even though still controversial) notion of most macro longitudinal studies, instead of S-allele carriers, L/L-allele carriers were the more susceptible group, which might reflect adaptive emotional flexibility”

Comment 4: The authors argue that 5-HTTLPR is of interest due to its associations to personality traits of negative emotionality. Have the authors assessed measures of personality? Normally one would expect that personality explains far more variance than a single genetic polymorphism. Explained variance < 2 %.

Response: We agree with the reviewer that personality traits could potentially explain more variance in differential susceptibility. The main purpose of the present paper was to investigate the role of individual differences in the serotonin system for differential susceptibility. There has been a substantial body of evidence which tested the differential susceptibility hypothesis of the serotonin system with the 5-HTTLPR. Based on this research, and candidate genes in general, the effect sizes are generally expected to be low. For this reason, we acquired a large sample (compared to non-genetic EMA studies) and prominently featured the issue of statistical power. Indeed, we see the 5-HTTLPR not as something that can explain a practically meaningful amount of variance, but rather as a tool to test hypotheses concerning neurotransmitter systems in humans with study designs which would otherwise not be feasible, such as EMA. 

We added this rational to the discussion:

“Last, although the effect sizes were reasonable for genetic studies on single polymorphisms, they are not large in absolute terms. Nevertheless, the main purpose of the present study was to test the theory that individual variability in the serotonin system is related to differential susceptibility. Searching for reliable effects of serotonergic genes is likely currently the only viable approach to test this theory in humans in daily life even if these effects are small.”

We agree that a study on the relationship between differential susceptibility and personality traits (e.g. big five) would be both relevant and interesting, but would merit a paper in its own right for two reasons:

1. There has been, to our knowledge, no attempt to develop a theory which systematically links trait measures like the big five to the concept of differential susceptibility. In the big five models of context sensitivity we are aware of, neuroticism is related to negative and extraversion to positive sensitivity (e.g. Larsen & Ketelaar, 1991). Past attempts to relate big five measures to differential susceptibility have not been successful (e.g. Slagt et al., 2015). Therefore, we think adding to this literature necessitates are more comprehensive paper based on a plausible theoretical framework.

2. A meta-analysis of large-scale genome-wide association studies reported no association between big five personality traits and serotonergic genes, including the 5-HTTLPR (Lot et al., 2017). Therefore, we think there is currently no sufficient basis to include personality measures in a paper that focuses on the 5-HTTLPR.

Larsen, R. J., & Ketelaar, T. (1991). Personality and susceptibility to positive and negative emotional states. Journal of personality and social psychology, 61(1), 132.

Slagt, M., Dubas, J. S., Denissen, J. J., Deković, M., & van Aken, M. A. (2015). Personality traits as potential susceptibility markers: Differential susceptibility to support among parents. Journal of Personality, 83(2), 155-166.

Lo, M. T., Hinds, D. A., Tung, J. Y., Franz, C., Fan, C. C., Wang, Y., ... & Sanyal, N. (2017). Genome-wide analyses for personality traits identify six genomic loci and show correlations with psychiatric disorders. Nature genetics, 49(1), 152.

Comment 5: The authors should in addition genotype for rs25531 which would allow a triallelic approach in evaluating their data. This is nowadays common standard in the analysis 5-HTTLPR.

Response: We would like to refrain from additional genotyping, for two reasons, and hope this is acceptable. The first one is practical; due to the current situation caused by the Corona virus, our laboratories have been shut, and it is unclear when we can resume our work. Second, although in vitro gene expression studies have shown that the G allele within the 5-HTTLPR l-allele was associated with lower 5HTT expression (Hu 2006), this finding has not been consistently replicated (Martin 2007). Furthermore, in other studies, examination of triallelic variation did not modify the results (e.g., Huezo-Diaz 2007, Steiger 2009, Wüst 2009).

Reviewer #2:

Comment 1: Abstract: I think the abstract should included some information on the sample and its size, (and maybe also on the analytic approach taken). Also, I suspect that the last sentence does not correspond to what the authors wanted to express (which is probably that the study provides insight into the serotonin system’s general role…).

Response: We thank the reviewer for the helpful remarks.

We added the italicized parts to the abstract: 

“Participants (N = 326, 77% female, mean age = 25, range = 17–36) completed smartphone questionnaires four times a day over four to five days”

“Affect was predicted from environment valence in the previous hour on a within-person level using three-level autoregressive linear mixed models.”

“Our study informs provides insight into the serotonin system’s general role in susceptibility and highlights the need to assess the whole spectrum of naturalistic experiences.”

Comment 2: In the sentence “Positive affect in carriers of the short allele…”, as well as in other parts of the manuscript, the authors make a comparison, but name only one comparison group (in this sentence the S-allele carriers), but not to which they are compared. I think these statements would gain clarity from complete descriptions of the comparisons.

Response: We added the reference group to the abstract and to several sentences in the manuscript.

Comment 3: P. 3, last sentence (minor): I think that the reference to the compatibility of the DS with the heritability patterns from twin studies needs a bit more explanation to fulfill its function here.

Response: We rewrote the sentence in the following way, to give more details:

“The differential susceptibility perspective is less well researched, despite simulations heritability coefficients simulated from differential susceptibility theory being compatible with empirical heritability coefficients usually observed in twin studies demonstrating its compatibility with heritability patterns of twin studies (Del Giudice, 2017).”

To give more information, we would need to discuss the simulation study in considerably more detail. If the reviewer finds our changes insufficient, we are also happy to remove this finding/reference. 

Comment 4: P4, 75ff. (minor): This second part of the paragraph is quite cursory and therefore not very informative and not very helpful.

Response: We rewrote the paragraph to make it more clear why we cite these findings. 

Comment 5: P4, 81ff.: When the 5-HTTLPR is introduced, it would be interesting if in a sentence or two, the authors would say something about the function of the region.

Response: We rewrote the paragraph, most importantly the following sentence:

“It consists of a 43 base pair insertion/deletion polymorphism in the promoter region of the serotonin transporter gene (SLC6A4) which influences the efficiency of serotonin reuptake from the synaptic cleft, with the L-variant leading to a more efficient reuptake through higher serotonin transporter gene transcription rates (Heils et al., 2002). “

Comment 6: P5, first paragraph: I think this paragraph needs some revision. The first observation the authors name does not support the hypothesis for the 5-HTTLPR to be a differential susceptibility factor. This first observation simply points to gap in research. It is not clear to me what the authors want to point to with the Sagy & Antonovsky 2000 reference. Then the second observation provides some ground for a DS hypothesis, but it is not the fact that most studies did not further examine this, that is informative. Finally, the “consequently” doesn’t really fit here to introduce the first partial sentence, and similarly, the second partial sentence doesn’t follow logically after the “but”.

Also related to this paragraph, I would find it important if the authors provided some insight into the literature and maybe a few sample studies that point to susceptibility of positive experience (e.g., Koenen’s work or Way at al., 2006).

Response: (a) In the first sentence of the paragraph, we did not want to imply that these two observations support the hypothesis, but rather explain what led to the conception of the hypothesis. Although these are just qualitative observations and not evidence for differential susceptibility, we included them as they are the essentials to understand differential susceptibility studies (1: inclusion of both positive and negative aspects; 2: cross-over interaction). We tried to make this clearer with a slight change: 

“The hypothesis that the 5-HTTLPR might actually be a susceptibility factor, and not merely a vulnerability factor, was mainly conceived based on two observations”

(b) The reference was demanded by a prior reviewer to illustrate a positive developmental effect. We don’t see a problem removing it and did so for this revision.

(c) We substituted “consequently” for “since then”. 

(d) We changed the “but”-sentence to: “Still, to date, the evidence is unclear as meta-analyses have not provided unequivocal evidence that the 5-HTTLPR is a differential susceptibility factor”

(e) We found the question whether additional single studies should be incorporated in this paragraph difficult, even before the submission. One the one hand, we agree with the reviewer that reading about specific studies can be helpful to make the topic more approachable. On the other hand, we had a hard time deciding which studies to pick, given that we already cite two meta-analyses which comprise a large number of both significant and null findings. As a possible compromise in the revision between the two options, we went into larger detail for the both meta-analysis to give readers more insight into what kind of studies have been done, without the possibility of us cherry-picking findings from certain areas/authors or significant vs null results:

“One meta-analysis found that negative developmental environments increased the likelihood of negative outcomes for carriers of the S-allele, compared to L/L-carriers, but did not find a robust overall effect of positive developmental environments (van Ijzendoorn, Belsky, & Bakermans-Kranenburg, 2012). Another meta-analysis based on randomized controlled trials did not show a significant difference in differential susceptibility for the 5-HTTLPR (van Ijzendoorn & Bakermans-Kranenburg, 2015). Interestingly, they observed that the pooled effect size of different hypothesized susceptibility genes depended on the timescale of measurements, with the largest effects for interventions which focus on immediate neural or behavioral responses to negative or positive stimuli)”

Comment 7: P5, 114ff.: I think this paragraph addresses a very important point and strength of this study. However, the last sentence, as formulated, may be correct but is not specific to the DS hypothesis, as it would apply as well to a vulnerability hypothesis.

Response: We completely agree with the reviewer and tried to make it clearer in the next paragraph:

“However, both differential susceptibility and the vulnerability to stressors are the property of a dynamic system that only manifests over time”

Comment 8: General point about EMA data and reactivity: This part has an implicit causal “touch”, and so it should be noted somewhere that we cannot draw conclusions about causality or a causal direction, since these are correlational data.

Response:

This is a very helpful remark. We added the following to the limitation section: 

“Third, although our study targets the process of environmental reactivity, which assumes a causal contingency between stimulus and output, our design is still correlational in nature. It is for example possible that momentary affect contaminates the ratings of previous stressors”

Comment 9: P6, 2nd paragraph, 130ff: The reference to the van Roekel et al study is interesting, but I think more detail is necessary to make this point with sufficient clarity. There is a long way from within-subject associations of daily experiences and affect, to AR(1)measures of emotional inertia, to the concept of emotional flexibility. The second sentence, at 132 ff., is unclear and maybe incomplete.

Response: We changed the paragraph to the following version. We hope this makes it clearer:

“In contrast, a recent EMA study reported that the S-allele was positively associated with affective inertia (van Roekel, Verhagen, Engels, & Kuppens, 2018), which is argued to reflect emotional inflexibility (van Roekel, Verhagen, Engels, & Kuppens, 2018) (Houben, Noortgate, & Kuppens, 2015). This finding conflicts with the hypothesis that S-allele induces higher differential susceptibility to environmental influences. However, affective inertia was operationalized as the autocorrelation of emotional states over time. A recent study demonstrated that this indicator might add only limited information to person-means on measures of emotion, leading the authors to suggest the additional assessment of concrete events and contextual information (Dejonckheere.et al., 2019)”

Comment 10: Section “Participants”: This section suggests that an a priori power analysis has been conducted to determine the necessary sample size. Therefore, the authors should report the sample size determined to achieve adequate power here. They can still give the details on the power analysis later on.

Response:The relevant text in the participants-paragraph reads:

“To estimate the appropriate sample size for effect detection in EMA studies, prior data with a similar design are necessary (Bolger & Laurenceau, 2013). As such data was not available, sample size was based on other genetic EMA studies on the same effect (e.g. Gunthert et al., 2007). See the results subsection on power considerations for further details”

This section thus states that we did not conduct an a priori power analysis, because we didn’t have EMA data to base our power analysis on at the time and therefore relied for our sample size on previous studies. 

We added some results of the section “power considerations” up here (emphasizing on “post-hoc”) to hopefully make this clearer: 

“A simulated post-hoc power analysis based on our empirical data indicated a power of 61.70% to detect a previously reported effect size (Gunthert et al., 2007), with a positive predictive value of 92.0% assuming equal prior probabilities. Notably, this previous effect size was smaller than for another study (Conway et al., 2014) and is therefore more conservative.”

Comment 11: P8, 193ff (minor): Because the authors begin a new paragraph, it was not immediately clear to me that the EFA still referred to the PANAS.

Response: We changed it accordingly.

Comment 12: Daily events: Could the authors provide more detail on this measure? It would be useful to know the instruction, and specially the labels used with the visual analogue scale. I was a bit surprised to see that the occurrence of an event was measured on a 0-100 scale, given that the scale measures events (but as someone who does a lot of EMA research I see the advantage of this approach). My concern would be that this kind of measures also taps into mood, and might spuriously inflate correlations with panas measures.

Response: We added the upper anchor of the rating scales: (upper anchor: “trifft zu“ [“applies”]). There was no lower anchor. The items were formulated in a way that didn’t make further instructions necessary and are stated in the methods section.

We completely agree on the point concerning the measurement of stressors and think that this is a grossly underresearched aspect. We added the following to the limitations:

“Last, our operationalization of environmental events through a semi-continuous visual analogue scales raises the question whether these measures are confounded by momentary affect. Still, the alternative of using binary indicators for event occurrence has shortcomings as well. For example, a person could indicate interpersonal conflicts due to very minor or extremely severe events, which has vastly different implications. Consequently, high stress reactivity might reflect more severe events. Also, binary indicators might not be sufficiently sensitive to capture minor stressors, which we specifically wanted to include in our study.”

Comment 13: I was glad to see that the authors didn’t report an alpha for the daily events measure, and fully agree with their justification! Something that I was wondering about was whether this wouldn’t apply to the PANAS, too. Probably not when it is used to measure mood-like affective states without particular emotional experiences occurring. Given the consistency scores reported, this seems to be the case here. This might be relevant to the interpretation of the results, since we’re probably not dealing with distinct emotional responses here, but rather with moderate shifts in participants mood.

Response:

To us, this appears to be a very complex question. We would argue that the existence of a latent construct like negative affect must be (1) supported by the data but still (2) theoretically imposed. Generally, there is likely no statistical way to tell whether there is latent “super construct” like negative affect or whether negative emotions are distinct but highly correlated constructs. There certainly are other CFA studies which consider affective states in EMA to be a latent construct (e.g. Rush & Hofer, 2014). We agree that EMA likely captures mood-like affective states. Whether the assumption of a latent affect construct for short/transient emotional states is valid is a question that we unfortunately cannot answer and might be beyond the scope of the paper. 

Rush, J., & Hofer, S. M. (2014). Differences in within- and between-person factor structure of positive and negative affect: Analysis of two intensive measurement studies using multilevel structural equation modeling. Psychological Assessment, 26(2), 462–473. doi: 10.1037/a0035666

Comment 14: P10, 224: Could the authors say more about the lack of power they refer here?

Response:

We thought this was the case because of the fewer degrees of freedom the nlme() output reports for between-subject effects, compared to cross-level interactions. Revisiting the power analyses tables of rush and hofer (2014), which shows similar power for the two effects, we don’t feel confident with this statement anymore and removed it.

Comment 15: P10 230: I think it would be useful if the authors provided their rationale for using these contrasts here. Why not contrasting both groups of S allele carriers from the homozygous L allele carriers?

Response:

We added the italicized parts to the paragraph:

“The trichotomous genotype was recoded into two Helmert contrasts in concordance with Gunthert et al. (2007): One contrast represents the mean difference between the L/L and the pooled S-carriers, with a positive sign indicating a larger association for the S-carriers. This contrast tests for the most common assumption that the S-allele is dominant. The other contrast represents the mean difference between the L/S- and the S/S-carriers, with a positive sign indicating a larger association for the S/S-carriers. This contrast tests for the additional possibility that the effect of the S-allele is additive (see Supplemental Methods 2 for details). In many previous studies, the comparison between the L/L-carriers and the carriers of at least one S-allele has been the central contrast of interest, while the contrast between L/S and S/S carriers is often not reported (van Ijzendoorn, Belsky & Bakermans-Kranenburg, 2012).”

Comment 16: P11: Was there an association between genotype and variability on affect and daily experience variables?

Response: We did not test this for this association, because Dejonckheere et al. (2019) demonstrated comprehensively (and in our opinion very convincingly) that indices of variability do not contain additional information above the person mean and the association between person means and genotype was clearly not significant (p > .80).

Dejonckheere, E., Mestdagh, M., Houben, M., Rutten, I., Sels, L., Kuppens, P., & Tuerlinckx, F. (2019). Complex affect dynamics add limited information to the prediction of psychological well-being. Nature Human Behaviour. doi: 10.1038/s41562-019-0555-0

Comment 17: P11, 262, minor: “Compared to” might fit better here than “instead”.

Response: We changed it accordingly

Comment 18: P13, 299-305 (minor): This paragraph is not very precise and therefore not easy to read (beginning: do you refer to one or more associations?). No difference on what? … more susceptible to what? From the context, one can figure out what you mean, but complete and more precise formulations would help.

Response: We changed the paragraph accordingly. Changes are italicized:

“The 5-HTTLPR moderated the within-person association between positive affect and both stressors and uplifts (Table 2). The association was strongest for L/L-carriers: Their The positive affect of L/L carriers was significantly more positively related to uplifts and more negatively related to stressors compared to S-carriers (Figure 2). There was no further reliable difference between L/S- and S/S-carriers in the association between positive affect and stressors or uplifts (Table 2). 5-HTTLPR genotype explained 5.76% and 4.73% of variance in reactivity slopes, respectively. In sum, these results corroborate that the 5-HTTLPR is a differential susceptibility factor, with L/L-carriers being more susceptible to environmental influences than carriers of the S-allele”

Comment 19: Results, general point: The effects, particularly that of testing differential susceptibility, is rather small. Did the authors expect an effect of this size, and do they consider it relevant nevertheless? I think it would be important that the authors commented on the effect sizes.

Response: We added the following to the limitations:

“Last, although the effect sizes were reasonable for genetic studies on single polymorphisms, they are not large in absolute terms. Nevertheless, the main purpose of the present study was to test the theory that individual variability in the serotonin system is related to differential susceptibility. Searching for reliable effects of serotonergic genes is likely currently the only viable approach to test this theory in humans in daily life even if these effects are small”

Comment 20: 

The discussion is relatively brief, if the piece on false positives is not counted. I would have hoped for a more elaborate discussion of the specifics of what the authors have actually studied.

The study period was quite brief, and similar studies collecting binary indications of daily events have occurred find relatively few such events, particularly negative ones, with a sizable portion of the sample report no negative events at all within a 4- or 5-day period. Are the authors convinced that all participants experienced clearly positive and negative events during the 4 or 5 days? This may point to the possibility that what was studied here was not necessarily emotional reactivity to events, in the sense of significant responses with an adaptive goal, but rather mild fluctuations of positivity as a function of relatively ordinary experiences. Some situations may be evaluated as more or less positive or negative, without representing a challenge or opportunity that would deserve a specific response. The average of uplifts and stressors seem clearly above 0, but given that participants probably reported with a finger on a smartphone display (not sure that this is how it was done), it may still be that much of the ratings at the lower 10-20% of the scale range may represent indications of no major event occurring. At least this is the experience we’re making with these kinds of tools.

All these possibilities may or may not have led to an attenuation of the “real” moderator effects. It could be interesting to see whether setting the threshold for registering an event to a value above the lowest 25% of the VAS would yield the same results.

Response: For the present study, we were indeed especially interested in minor events, which can happen on a day to day basis in different degrees. We added the following italicized passage to the introduction to make this clearer:

“EMA methods are capable of measuring even minor daily experiences which often have a higher predictive utility than major life events and are the main target in the present study”

Also, the following sentence was added to the limitations for a previous comments:

“Also, binary indicators might not be sufficiently sensitive to capture minor stressors, which we specifically aimed for in our study”

Including the text added due to the review comments, the discussion now has over 1600 words and is relatively long, compared with similar genetic papers. We do think that the discussion on false positives in candidate gene studies is an integral part, which should be counted. It addresses the most consistent and prominent response we received when presenting this research. 

Comment 21: P18, 390 (minor): Although the authors note that they expect stronger associations for s allele carriers in the intro, it was not too clear to me that they did have this expectation since it is not marked clearly as a hypothesis.

Response: We exchanged the word “expected” for “hypothesized” to make this clearer.

Comment 22: P19, top: the reference to the dynamic system is unclear. I suggest that the authors either provide more detail about it, or delete the reference. In the next sentence, it is not clear what levels the authors refer to. Generally, I think this sentence must be rewritten. I think I know what they authors mean, but the sentence lacks precision and sufficient information for the reader less familiar with the topic.

Response: We tried to make the sentence clearer:

“In contrast to most studies on the 5-HTTLPR, the main purpose of our study was to investigate environmental reactivity as a dynamic system and focus on a within-person effectslevel. Expecting within- and between-person that these two levels analyses must yield the same results constitutes a so-called ecological fallacy andas the necessary assumptions of this equality which equate the two levels are rarely met ”

Comment 23: General remark about the discussion: I find it important and interesting that the authors try to connect these results with other phenomena related to emotional adaptation and adjustment, and patterns of emotion patterns. However, this research did not examine flexibility or emotion patterns, and sometimes the authors risk to be overly speculative. Also, the cover a large number of different topics, each one hinting at some research and touching those only very superficially. Several parts of the discussion are therefore accessible only to the readers deeply familiar with the topic. Less (with more detail) would be more here.

Response: We think the reviewer makes an important point in that this ad-hoc hypothesis to explain the surprising results needed to express more caution. We added the following to the paragraph:

“Still, more research is needed to support this possible explanation as our study design is not sufficient to link environmental reactivity with adaptive emotional functioning”

Because it is only an ad hoc explanation which needs more research, we tried not to give it too much space in the discussion, but rather point it out as a possibility. We had several clinical colleagues without prior knowledge in the area read it and are now relatively confident that the main point is understandable to this group.

Comment 24: The following papers may be relevant to this study.

Schoebi, D., Way, B. M., Karney, B. R., & Bradbury, T. N. (2012). Genetic moderation of sensitivity to positive and negative affect in marriage. Emotion, 12(2), 208.

Haase, C. M., Saslow, L. R., Bloch, L., Saturn, S. R., Casey, J. J., Seider, B. H., ... & Levenson, R. W. (2013). The 5-HTTLPR polymorphism in the serotonin transporter gene moderates the association between emotional behavior and changes in marital satisfaction over time. Emotion, 13(6), 1068.

Response: We read both papers and think that particularly the first paper is informative for our study. We added it to the introduction.

---

## [Decision Letter · Decision Letter 1]

11 May 2020

PONE-D-20-03284R1

Highs and lows: Genetic susceptibility to daily events

PLOS ONE

Dear Mr. Sicorello,

Thank you for submitting your manuscript to PLOS ONE. After careful consideration, we feel that it has merit but does not fully meet PLOS ONE’s publication criteria as it currently stands. Therefore, we invite you to submit a revised version of the manuscript that addresses the points raised during the review process.

Thank you for your careful revision of your manuscript, taking most points of the reviewers into account.

As you can see below, the two reviewers have some remaining points and one of them raises that your revision was not yet satisfactory in addressing their concerns.

I invite you to revise your manuscript addressing specifically the points of reviewer #1 on referring to literature on the relationship between personality and differential susceptibility and clarifying your aim regarding personality versus individual differences in susceptibility; regarding point (3), the additional analyses for the genotype rs25531: if you cannot do those analyses in the near future due to closed labs, I recommend you explaining that in your manuscript, if you can do those analyses within the next weeks, you should add the results to the manuscript.

Regarding reviewer #2's comments: the point about a-priori power analysis and post-hoc power calculation is well made, using another publication's effect size to calculate an a-priori sample size would be appropriate, so this should be amended in the manuscript; also, some clarification around the wording for dynamic systems/reactivity would be helpful. 

We would appreciate receiving your revised manuscript by Jun 25 2020 11:59PM. To enhance the reproducibility of your results, we recommend that if applicable you deposit your laboratory protocols in protocols.io, where a protocol can be assigned its own identifier (DOI) such that it can be cited independently in the future. For instructions see: http://journals.plos.org/plosone/s/submission-guidelines#loc-laboratory-protocols

We look forward to receiving your revised manuscript.

Kind regards,

Hedwig Eisenbarth

Academic Editor

PLOS ONE

Additional Editor Comments (if provided):

Thank you for your careful revision of your manuscript, taking most points of the reviewers into account. Your revised version has improved significantly, however, there are some remaining points that need to be addressed.

Reviewers' comments:

Reviewer's Responses to Questions

**Comments to the Author**

1. If the authors have adequately addressed your comments raised in a previous round of review and you feel that this manuscript is now acceptable for publication, you may indicate that here to bypass the “Comments to the Author” section, enter your conflict of interest statement in the “Confidential to Editor” section, and submit your "Accept" recommendation.

Reviewer #1: All comments have been addressed

Reviewer #2: (No Response)

2. Is the manuscript technically sound, and do the data support the conclusions?

Reviewer #1: Yes

Reviewer #2: Yes

3. Has the statistical analysis been performed appropriately and rigorously? 

Reviewer #1: Yes

Reviewer #2: Yes

4. Have the authors made all data underlying the findings in their manuscript fully available?

Reviewer #1: Yes

Reviewer #2: Yes

5. Is the manuscript presented in an intelligible fashion and written in standard English?

Reviewer #1: Yes

Reviewer #2: Yes

6. Review Comments to the Author

Reviewer #1: 1. There has been, to our knowledge, no attempt to develop a theory which systematically links trait measures like the big five to the concept of differential susceptibility. In the big five models of context sensitivity we are aware of, neuroticism is related to negative and extraversion to positive sensitivity (e.g. Larsen & Ketelaar, 1991). Past attempts to relate big five measures to differential susceptibility have not been successful (e.g. Slagt et al., 2015). Therefore, we think adding to this literature necessitates are more comprehensive paper based on a plausible theoretical framework.

There is abundance of literature showing that personality traits are related to differential susceptibility. No matter what the authors mean with “differential susceptibility” if they mean variability in mood (states) or susceptibility to psychopathology. Neuroticism is defined a s a trait related to the predisposition to be “moody” and there is clear proneness for anxiety or affective disorders (e.g. De Moor et al., 2015).

2. A meta-analysis of large-scale genome-wide association studies reported no association between big five personality traits and serotonergic genes, including the 5-HTTLPR (Lot et al., 2017). Therefore, we think there is currently no sufficient basis to include personality measures in a paper that focuses on the 5-HTTLPR.

Bad argument, sorry. You are not aiming to demonstrate an association between 5-HTTLPR and personality. You aim to explain variance in susceptibility. If predictors are uncorrelated this is a clear advantage for explaining additional variance in your dependent variable.

3. The authors should in addition genotype for rs25531 which would allow a triallelic approach in evaluating their data. This is nowadays common standard in the analysis 5-HTTLPR.

Response: We would like to refrain from additional genotyping, for two reasons, and hope this is acceptable. The first one is practical; due to the current situation caused by the Corona virus, our laboratories have been shut, and it is unclear when we can resume our work. Second, although in vitro gene expression studies have shown that the G allele within the 5-HTTLPR l-allele was associated with lower 5HTT expression (Hu 2006), this finding has not been consistently replicated (Martin 2007). Furthermore, in other studies, examination of triallelic variation did not modify the results (e.g., Huezo-Diaz 2007, Steiger 2009, Wüst 2009).

As I already said, the triallelic approach is common standard when genotyping for 5-HTTLPR. Technically, you can run this in the same analysis, with the exception with an enzymatic digestion in-between. I am confident that the laboratories will open soon. And one person alone can conduct the analyses safely alone in the lab.

Reviewer #2: I think the authors did a great job responding to my comments.

I only have two additional comments, which the authors may consider:

1. Regarding (a priori) power analysis (or not)/ comment 10: I my first review, I misread the relevant section, and I think this was for a good reason (or logic). If there were prior genetic studies on the same effect (Gunthert et al), then why not using those data for an a priori power analysis - just as the authors seem to have done post hoc? That would seem possible to me, and that's why I mistakenly thought an a priori power analysis has been done. Besides, if not for using information of a prior study for a power analysis, I don't see how a prior study can inform about what sample size to aim for.

2. (referring to my comment 22): Regarding this and some of my other comments on the discussion, the authors seem to have a slightly different opinion, in some instances, on where and how to set the emphasis, which is fair enough and totally ok for me. The one minor point where I think it is worth giving it another thought is their reference to investigate reactivity as a dynamic system. I totally agree about the importance of testing such effects on the within-subject level. But I'm not sure it is clear what they investigated as a dynamic system. Reactivity is dynamic per definition, so this sentence seems a little tautological.

7. PLOS authors have the option to publish the peer review history of their article (what does this mean?). If published, this will include your full peer review and any attached files.

Reviewer #1: No

Reviewer #2: Yes: Dominik Schöbi

---

## [Author Response · Author response to Decision Letter 1]

19 May 2020

Reviewer #1:

Comment 1: There is abundance of literature showing that personality traits are related to differential susceptibility. No matter what the authors mean with “differential susceptibility” if they mean variability in mood (states) or susceptibility to psychopathology. Neuroticism is defined a s a trait related to the predisposition to be “moody” and there is clear proneness for anxiety or affective disorders (e.g. De Moor et al., 2015).

Response:

We do not share the Reviewer’s assessment of an abundant literature linking neuroticism to differential susceptibility. „Susceptibility to psychopathology“ reflects the concept of vulnerability, which is distinct from differential susceptibility. There is a sound theoretical basis for the differential susceptibility concept (Belsky, 1997; Belsky et al., 2007; Pluess, 2015), and this theory represents the basis of our study. As outlined in the introduction of our paper, the concept of differential susceptibility was developed to show there are different patterns of reactivity to different environmental circumstances, which go beyond the concept of vulnerability. Differential susceptibility addresses predisposing factors which increase the susceptibility to both negative and positive events. While neuroticism surely is positively related to the vulnerability to negative events (Ormel et al., 2013; Shackman et al., 2016), we are not aware of compelling evidence showing that neuroticism also leads individuals to profit more from positive events, which is -the core of differential susceptibility theory. For example, individuals high in neuroticism show lower increases of positive affect in experimental inductions of positive mood (Larsen & Ketelaar, 1991), which is in agreement with Gray’s (1981) neuroticism model. Specifically in the EMA context, the negative affect of individuals high in neuroticism is more sensitive to negative, but not positive events (for a well-powered example see Longua et al., 2009).

Belsky, J. (1997). Theory Testing, Effect-Size Evaluation, and Differential Susceptibility to Rearing Influence: The Case of Mothering and Attachment. Child Development, 68(4), 598. https://doi.org/10.2307/1132110

Belsky, J., Bakermans-Kranenburg, M. J., & van IJzendoorn, M. H. (2007). For better and for worse: differential susceptibility to environmental influences. Current Directions in Psychological Science, 16(6), 300–304. https://doi.org/10.1111/j.1467-8721.2007.00525.x

Gray, J. A. (1981). A critique of Eysenck's theory of personality. In H. J. Eysenck (Ed.), A model for personality (pp. 246-276). New York: Springer.

Larsen, R. J., & Ketelaar, T. (1991). Personality and Susceptibility to Positive and Negative Emotional States. Journal of Personality and Social Psychology, 61(1), 132–140. https://doi.org/10.1037/0022-3514.61.1.132

Longua, J., DeHart, T., Tennen, H., & Armeli, S. (2009). Personality moderates the interaction between positive and negative daily events predicting negative affect and stress. Journal of Research in Personality, 43(4), 547–555. https://doi.org/10.1016/j.jrp.2009.02.006

Ormel, J., Jeronimus, B. F., Kotov, R., Riese, H., Bos, E. H., Hankin, B., Rosmalen, J. G. M., & Oldehinkel, A. J. (2013). Neuroticism and common mental disorders: Meaning and utility of a complex relationship. Clinical Psychology Review, 33(5), 686–697. https://doi.org/10.1016/j.cpr.2013.04.003

Pluess, M. (2015). Individual Differences in Environmental Sensitivity. Child Development Perspectives, 9(3), 138–143. https://doi.org/10.1111/cdep.12120

Shackman, A. J., Tromp, D. P. M., Stockbridge, M. D., Kaplan, C. M., Tillman, R. M., & Fox, A. S. (2016). Dispositional negativity: An integrative psychological and neurobiological perspective. Psychological Bulletin, 142(12), 1275–1314. https://doi.org/10.1037/bul0000073

Comment 2: Bad argument, sorry. You are not aiming to demonstrate an association between 5-HTTLPR and personality. You aim to explain variance in susceptibility. If predictors are uncorrelated this is a clear advantage for explaining additional variance in your dependent variable.

(referring to our previous response: “A meta-analysis of large-scale genome-wide association studies reported no association between big five personality traits and serotonergic genes, including the 5-HTTLPR (Lot et al., 2017). Therefore, we think there is currently no sufficient basis to include personality measures in a paper that focuses on the 5-HTTLPR.)” 

Response:

We assumed the reviewer was suggesting that neuroticism might mediate the effect of 5-HTTLPR on differential susceptibility. We agree, as the reviewer was not suggesting this relationship, our argument of no association between serotonergic genes and personality does not apply.

The aim of our study was to test the hypothesis that specifically the 5-HTTLPR is related to individual differences in differential susceptibility. As noted in the introduction and discussion, this hypothesis is embedded in the larger theory of the role of the serotonin system. It was not the aim of our study to find predictors which explain large amounts of variance in differential susceptibility. Single polymorphisms are known to explain only small amounts of variance, which are not practically meaningful for prediction, further highlighting the inconsistency between these two aims. We added the following phrase in response to a comment of the previous round of reviews, which makes this explicit:

“Last, although the effect sizes were reasonable for genetic studies on single polymorphisms, they are not large in absolute terms. Nevertheless, the main purpose of the present study was to test the theory that individual variability in the serotonin system is related to differential susceptibility”

Taken together, we still believe there is an insufficient basis to include a predictor of no interest (i.e. neuroticism) which is empirically unrelated to our predictor of interest (i.e. 5-HTTLPR) and has no clear straight-forward relationship to the outcome of interest (i.e. differential susceptibility; see response to comment 1).

Comment 3: 

As I already said, the triallelic approach is common standard when genotyping for 5-HTTLPR. Technically, you can run this in the same analysis, with the exception with an enzymatic digestion in-between. I am confident that the laboratories will open soon. And one person alone can conduct the analyses safely alone in the lab.

Response: Unfortunately, we are currently still in no position to repeat those analyses. We added the following paragraph to the discussion: 

„A further limitation is that we did not genotype1 for rs25531 A/G SNP within the 5-HTTLPR repetitive element, which has been reported to render the G allele within the l allele transcriptionally less efﬁcient (Wendland 2006), a finding that has not been consistently replicated (Martin 2007)”

1Footnote: Reanalysis of the available DNA samples was not possible due to Corona-associated lab closure. 

Reviewer #2: I think the authors did a great job responding to my comments.

I only have two additional comments, which the authors may consider:

Comment 1.1: Regarding (a priori) power analysis (or not)/ comment 10: I my first review, I misread the relevant section, and I think this was for a good reason (or logic). If there were prior genetic studies on the same effect (Gunthert et al), then why not using those data for an a priori power analysis - just as the authors seem to have done post hoc? That would seem possible to me, and that's why I mistakenly thought an a priori power analysis has been done. 

Response:

First, we changed the following sentence to clarify that we did not conduct an a priori power analysis before data collection to determine the sample size (changes in italics): “As such data was not available, our sample size was based on the sample size of other genetic EMA studies on the same effect (e.g. [33]).” 

We also started a new paragraph after this sentence, to highlight the difference between sample size decision and post hoc power analysis. 

For power estimation in linear mixed models, information are necessary on both (a) the expected effect size and (b) the expected covariance structure. While we were able to calculate a standardized effect size from Gunthert et al., the extraction of the covariance structure would not have been possible with the information given in the paper and would likely not be meaningful due to differences in the sampling plan. We tried to express this with the following sentence:

“To estimate the appropriate sample size for effect detection in EMA studies, prior data with a similar design are necessary [39]. As such data was not available, sample size was based on other genetic EMA studies on the same effect (e.g. [33])” [method section]

Therefore, we combined the previous effect size by Gunthert et al. with the empirical covariance structure of our design. We tried to express this more specifically in the results section (which we refer to in the method section on power for further detail):

“we tested the statistical power of our design based on our empirical covariance structure and the effect sizes reported by Gunthert et al. [33]. […] We followed the procedure described in Green and Macleod [49]” [results section]

Notably, in hindsight, it would have been possible to simulate a range of plausible true covariance structures and determine a power range before data collection. Unfortunately, we did not consider this approach at the time of study planning.

Comment 1.2: Besides, if not for using information of a prior study for a power analysis, I don't see how a prior study can inform about what sample size to aim for.

Response:

We wrote: “A simulated post-hoc power analysis based on our empirical data indicated a power of 61.70% to detect a previously reported effect size [33]”

The term “post-hoc power analysis”, as we are familiar with it, refers to the estimation of power, given that all other relevant parameters are fixed (i.e. sample size, alpha level, effect size, and covariance structure) and therefore does not provide a means of sample size estimation, but rather a means to indicate what the power of a given study has to detect an effect size of interest.

This is substantially inferior to an a priori power analysis for sample size estimation, but we included it nevertheless, as prior reviewers consistently expressed the need to give some indication of achieved statistical power for reasonable effect sizes, due to the ongoing discussion concerning limited power in candidate gene studies. 

2. (referring to my comment 22): Regarding this and some of my other comments on the discussion, the authors seem to have a slightly different opinion, in some instances, on where and how to set the emphasis, which is fair enough and totally ok for me. The one minor point where I think it is worth giving it another thought is their reference to investigate reactivity as a dynamic system. I totally agree about the importance of testing such effects on the within-subject level. But I'm not sure it is clear what they investigated as a dynamic system. Reactivity is dynamic per definition, so this sentence seems a little tautological.

If we are correct, the phrase the reviewer points towards is in the introduction (we hope we didn’t overlook a similar phrase in the discussion): “However, both differential susceptibility and the vulnerability to stressors are the property of a dynamic system that only manifests over time”. 

We completely agree that reactivity is dynamic per definition. This is what we try to express with this sentence. Unfortunately, most studies on differential susceptibility relied on cross-sectional between-person correlations to still draw conclusions for reactivity. We tried to highlight this in the phrase before the dynamic system reference: “Inconsistent findings in this area might be partly due to the fact that the overwhelming majority of studies attained cross-sectional data, and correlations were usually computed on a between-person level (e.g. [18])”

Thus, while we ourselves entirely agree with you and find this statement to be self-evident, we included it because we are under the impression this is not clear to most parts of the literature which our study is based on.

If the reviewer finds this explanation insufficient, we are of course open for further comments and suggestions.

---

## [Decision Letter · Decision Letter 2]

4 Jul 2020

PONE-D-20-03284R2

Highs and lows: Genetic susceptibility to daily events

PLOS ONE

Dear Dr. Sicorello,

Thank you for submitting your manuscript to PLOS ONE. After careful consideration, we feel that it has merit but does not fully meet PLOS ONE’s publication criteria as it currently stands. Therefore, we invite you to submit a revised version of the manuscript that addresses the points raised during the review process.

I'm very sorry for the delay of this process, but I wanted to get an additional view as the two reviewers differed in their view. Can I please ask you to address/answer the two questions of reviewer #3, please?

We look forward to receiving your revised manuscript.

Kind regards,

Hedwig Eisenbarth

Academic Editor

PLOS ONE

Reviewers' comments:

Reviewer's Responses to Questions

**Comments to the Author**

1. If the authors have adequately addressed your comments raised in a previous round of review and you feel that this manuscript is now acceptable for publication, you may indicate that here to bypass the “Comments to the Author” section, enter your conflict of interest statement in the “Confidential to Editor” section, and submit your "Accept" recommendation.

Reviewer #2: All comments have been addressed

Reviewer #3: (No Response)

2. Is the manuscript technically sound, and do the data support the conclusions?

Reviewer #2: Yes

Reviewer #3: Yes

3. Has the statistical analysis been performed appropriately and rigorously? 

Reviewer #2: Yes

Reviewer #3: Yes

4. Have the authors made all data underlying the findings in their manuscript fully available?

Reviewer #2: Yes

Reviewer #3: Yes

5. Is the manuscript presented in an intelligible fashion and written in standard English?

Reviewer #2: Yes

Reviewer #3: Yes

6. Review Comments to the Author

Reviewer #2: I think the authors addressed my earlier comments appropriately. I thought my fellow reviewer pointed out other potentially important issues.

Reviewer #3: I was asked to review this manuscript, while not being involved in the previous revisions. I honestly have to say I started out skeptical, due to the current state of the field with regard to candidate gene studies. However, I was pleasantly surprised. I think the authors did a great job in rigorously testing the hypotheses, and were very nuanced and complete in interpreting their findings. This included discussing the limitations, but also the controversies surrounding candidate gene studies.

I was a bit hesitant with regard to the difference scores that were used to test differential susceptibility. I think these scores have many problems, as the authors also state in their Discussion. The analyses split for events and affect provide much more insight into the mechanisms. I am also not convinced that averaging the event scores is the best solution, as certain events may be more impactful than others. However, given the state of the manuscript and potential issues with multiple testing, I will not ask for different analyses.

I have two remarks/questions:

(1) the authors use a three-level model (moment, day, person); did they check whether the second (day) level was required? I could not find whether the study period included week days only or also weekend days.

(2) I agree with reviewer 1 that the tri-allelic approach is preferable, so if it would be possible, this would be nice to add. However, I can also imagine that labs are still not working on full speed currently.

Finally, I would like to stress that I think that pre-registration and direct replication is essential in these studies, and I would advise the authors to do this next time. Particularly in this field, pre-registration may help to solve the inconsistencies in previous work.

7. PLOS authors have the option to publish the peer review history of their article (what does this mean?). If published, this will include your full peer review and any attached files.

Reviewer #2: No

Reviewer #3: **Yes: **Eeske van Roekel

---

## [Author Response · Author response to Decision Letter 2]

13 Jul 2020

Reviewer #3: I was asked to review this manuscript, while not being involved in the previous revisions. I honestly have to say I started out skeptical, due to the current state of the field with regard to candidate gene studies. However, I was pleasantly surprised. I think the authors did a great job in rigorously testing the hypotheses, and were very nuanced and complete in interpreting their findings. This included discussing the limitations, but also the controversies surrounding candidate gene studies.

I was a bit hesitant with regard to the difference scores that were used to test differential susceptibility. I think these scores have many problems, as the authors also state in their Discussion. The analyses split for events and affect provide much more insight into the mechanisms. I am also not convinced that averaging the event scores is the best solution, as certain events may be more impactful than others. However, given the state of the manuscript and potential issues with multiple testing, I will not ask for different analyses.

Response:

We thank the Reviewer for her critical and at the same time validating comments concerning our manuscript. We will respond to the individual comments below:

Comment 1

the authors use a three-level model (moment, day, person); did they check whether the second (day) level was required? I could not find whether the study period included week days only or also weekend days.

Response:

We added the following information to the method section: 

“Assessments always started on a Thursday to sample both working days and weekend days in a similar frequency”

“A likelihood ratio test for an intercept-only model with mood as the dependent variable indicated that including the day level increased model fit significantly (χ²(1) = 55.34, p < .001; 46.6% variance between participants, 11.7% between days, and 47.7% between measurement occasions)”

Comment 2

I agree with reviewer 1 that the tri-allelic approach is preferable, so if it would be possible, this would be nice to add. However, I can also imagine that labs are still not working on full speed currently.

Response: 

There are two issues regarding genotyping of rs25531. First, as the reviewer pointed out, our University is still running on a limited scale and our lab is currently allowed to work with 50% capacity only. There is still a major backlog of time-critically funded studies. Therefore, the re-analysis of this sample could unfortunately not be accomplished in the foreseeable future.

The second reason are doubts surrounding the effects of rs25531 on 5HTT expression. In 2007, Martin et al. comprehensively analyzed 55 SNPs in 100 kb surrounding the SLC6A4 gene regarding their effect on gene expression. In addition to the VNTR („5HTTLPR“), two other SNPs (both with low minor allele frequency) had a significant effect, but not rs25531.

 <Figure from Martin et al. (2007) is shown in the submitted document "ResponseToReviewers". Caption: Mapping regulatory variants for the serotonin transporter gene based on allelic expression imbalance. Mol Psychiatry.

We hope these reasons, as well as their acknowledgment in the limitations section, are acceptable.

Comment 3:

Finally, I would like to stress that I think that pre-registration and direct replication is essential in these studies, and I would advise the authors to do this next time. Particularly in this field, pre-registration may help to solve the inconsistencies in previous work.

Response:

We entirely agree with the reviewers’ comment and are continuously working to improve our reproducibility and replicability practices.

---

## [Decision Letter · Decision Letter 3]

20 Jul 2020

Highs and lows: Genetic susceptibility to daily events

PONE-D-20-03284R3

Dear Dr. Sicorello,

We’re pleased to inform you that your manuscript has been judged scientifically suitable for publication and will be formally accepted for publication once it meets all outstanding technical requirements.

Kind regards,

Hedwig Eisenbarth

Academic Editor

PLOS ONE

Additional Editor Comments (optional):

Reviewers' comments:

Reviewer's Responses to Questions

**Comments to the Author**

1. If the authors have adequately addressed your comments raised in a previous round of review and you feel that this manuscript is now acceptable for publication, you may indicate that here to bypass the “Comments to the Author” section, enter your conflict of interest statement in the “Confidential to Editor” section, and submit your "Accept" recommendation.

Reviewer #3: All comments have been addressed

2. Is the manuscript technically sound, and do the data support the conclusions?

Reviewer #3: Yes

3. Has the statistical analysis been performed appropriately and rigorously? 

Reviewer #3: Yes

4. Have the authors made all data underlying the findings in their manuscript fully available?

Reviewer #3: Yes

5. Is the manuscript presented in an intelligible fashion and written in standard English?

Reviewer #3: Yes

6. Review Comments to the Author

Reviewer #3: (No Response)

7. PLOS authors have the option to publish the peer review history of their article (what does this mean?). If published, this will include your full peer review and any attached files.

Reviewer #3: **Yes: **Eeske van Roekel

---

## [Editor Report · Acceptance letter]

23 Jul 2020

PONE-D-20-03284R3 

Highs and lows: Genetic susceptibility to daily events 

Dear Dr. Sicorello:

I'm pleased to inform you that your manuscript has been deemed suitable for publication in PLOS ONE. Congratulations! Your manuscript is now with our production department. 

Kind regards, 

on behalf of

Dr. Hedwig Eisenbarth 

Academic Editor

PLOS ONE